# Role of Liquid Biopsy for Early Detection, Prognosis, and Therapeutic Monitoring of Hepatocellular Carcinoma

**DOI:** 10.3390/diagnostics15131655

**Published:** 2025-06-28

**Authors:** Faris Alrumaihi

**Affiliations:** Department of Medical Laboratories, College of Applied Medical Sciences, Qassim University, Buraydah 51452, Saudi Arabia; f_alrumaihi@qu.edu.sa; Tel.: +966-555181862

**Keywords:** liquid biopsy, hepatocellular carcinoma, circulating tumor cells, cell free nucleic acids, prognosis

## Abstract

The global prevalence of hepatocellular carcinoma (HCC) is getting worse, leading to an urgent need for improved diagnostic and prognostic strategies. Liquid biopsy, which analyzes circulating tumor cells (CTCs), cell-free DNA (cfDNA), cell-free RNA (cfRNA), and extracellular vesicles (EVs), has emerged as a minimally invasive and promising alternative to traditional tissue biopsy. These biomarkers can be detected using sensitive molecular techniques such as digital PCR, quantitative PCR, methylation-specific assays, immunoaffinity-based CTC isolation, nanoparticle tracking analysis, ELISA, next-generation sequencing, whole-genome sequencing, and whole-exome sequencing. Despite several advantages, liquid biopsy still has challenges like sensitivity, cost-effectiveness, and clinical accessibility. Reports highlight the significance of multi-analyte liquid biopsy panels in enhancing diagnostic sensitivity and specificity. This approach offers a more comprehensive molecular profile of HCC, early detection, and tracking therapeutic treatment, particularly in those cases where single-analyte assays and imaging fail. The technological advancement in the isolation and analysis of CTC, cell-free nucleic acids, and EVs is increasing our understanding of extracting genetic information from HCC tumors and discovering mechanisms of therapeutic resistance. Furthermore, crucial information on tumor-specific transcriptomic and genomic changes can be obtained using cfRNA and cfDNA released into the peripheral blood by tumor cells. This review provides an overview of current liquid biopsy strategies in HCC and their use for early detection, prognosis, and monitoring the effectiveness of HCC therapy.

## 1. Introduction

Cancer is one of the leading causes of death worldwide, and it accounts for approximately 7.6 million deaths annually. This figure is expected to rise to 13.1 million by 2030 [1]. Current treatment strategies, such as chemotherapy and radiation therapy, are often linked to a number of adverse health effects [2]. Numerous risk factors, including diet, tobacco and alcohol use, infections, certain medications, and exposure to environmental hazards such as chemicals, industrial effluents, and ionizing radiation, influence the initiation and development of cancer. Additionally, genetic and epigenetic alterations have a significant role in carcinogenesis. These factors may act synergistically or independently to initiate or accelerate the progression of cancer. These risk factors are broadly classified into intrinsic and non-intrinsic risk factors [3]. The disruption of balance between reactive oxygen species (ROS) production in cancer leads to DNA damage and genomic instability [4]. The genetic landscape of cancer cells varies greatly among patients, and the physiological states of tumors are influenced by numerous factors, including drug-induced selection pressures, surrounding microenvironmental variables, and metabolic homeostatic systems within the tumor microenvironment. Advanced knowledge of tumor heterogeneity has emerged due to recent technical developments in multi-omics platforms. The intratumoral and interpatient heterogeneity poses a great issue in effective cancer treatment and can be considered an important contributor to treatment failure [5].

Early detection of cancer can be an important goal for effective cancer treatment, and it can screen the possible malignancies before the onset of symptoms, lower the risk of mortality due to cancer disease, and offer more potential advantages than disadvantages [6]. Conventional cancer diagnostic techniques, such as positron emission tomography (PET), X-ray computed tomography (CT), magnetic resonance spectroscopy (MRS), and molecular diagnostic techniques, have been documented to be crucial for both early cancer detection and the clinical management of cancer patients [7].

The early detection of cancer can be an important goal for effective cancer treatment, and it can screen for possible malignancies before the onset of symptoms, lower the risk of mortality due to cancer disease, and offer more potential advantages than disadvantages [6]. Conventional cancer diagnostic techniques, such as positron emission tomography (PET), X-ray computed tomography (CT), magnetic resonance spectroscopy (MRS), and molecular diagnostic techniques, have been documented to be crucial for both early cancer detection and the clinical management of cancer patients [7].

Tissue biopsy is a gold standard procedure for cancer diagnosis, and it involves the removal of a piece of tissue or a sample of cells using a needle or by surgery. Basically, tissue biopsy can be incisional biopsy, excisional biopsy, punch biopsy, shave biopsy, curettage biopsy, needle biopsy, imprint biopsy, etc. After the removal of tissue, it is sent for histopathological analysis [8]. Despite some evidence supporting their diagnostic efficacy, biopsies do not produce conclusive results. Biopsies are helpful for examining primary lesions of metastatic soft tissue and bone malignancies as well as for early diagnosis. In general, biopsies are helpful in determining the primary site of bone and soft tissue metastases as well as in diagnosing them. Consequently, by facilitating timely diagnosis and suitable therapy, tumor biopsies (in conjunction with strong interdepartmental cooperation) can enhance patient outcomes [9]. Traditional biopsy has several significant limitations. For example, it can be difficult to obtain sufficient tumor tissue, especially metastasis. Further, this approach is invasive, risky, and painful [10]. In addition to tissue biopsy, imaging techniques are frequently used for liver cancer diagnosis. However, their use is still limited due to several challenges, including suboptimal sensitivity, etc. [11]. Recognizing these challenges associated with tissue biopsy and imaging techniques, current research has shifted to liquid biopsy, which has several advantages [10,12,13].

The liver is one of the most important vital organs of the human body that plays a crucial role in maintaining metabolic homeostasis in the body [14]. The liver plays a multifaceted role in physiological processes such as digestion, xenobiotic detoxification, gluconeogenesis, lipid metabolism and storage, iron homeostasis, and the synthesis of plasma proteins. The liver also converts ammonia to urea, processes bilirubin, and breaks down hemoglobin [15]. It produces bile needed for breaking down fats. The liver also detoxifies medications and other toxic compounds. Additionally, it fights infections. Hence, liver diseases are important factors of morbidity and mortality. They may rise due to viral infections, exposure to chemicals and drugs, obesity, diabetes, and autoimmune disorders. Untreated liver diseases can result in chronic liver diseases, cirrhosis, or hepatocellular carcinoma (HCC), which can ultimately lead to life-threatening issues and death [16].

HCC accounts for almost 90% of primary liver cancers, followed by intrahepatic cholangiocarcinoma and other less common subtypes [17]. Liver cancer is the second most common cause of cancer-related death for males worldwide. Liver cirrhosis and HCC are significantly influenced by multiple factors, including genetic differences, gut microbial dysbiosis, aging, obesity, gender, and alcohol consumption. Additionally, chronic viral infections, including hepatitis B virus (HBV) and hepatitis C virus (HCV), are well-established drivers of liver carcinogenesis, while hepatitis D virus (HDV) accelerates advanced liver disease and its progression in populations having HBV infection [18].

Given the global burden of liver cancer and associated challenges of detection and management, liquid biopsy has emerged as a promising, less invasive technique for such purposes. This technique uses tumor-derived biomaterials such as circulating tumor cells (CTCs), cell-free DNA (cfDNA), cell-free RNA (cfRNA), and extracellular vesicles (EVs) in bodily fluids such as blood, saliva, breast milk, and urine. These components can be released by cancer cells and the stromal cells of tumor tissues. Especially, this release occurs during the metastatic phase, involving the detachment of cancer cells from the primary tumor tissue and entrance into their circulation through the bloodstream. These circulating substances have been documented to be biomarkers of pathogenic processes, normal biological activities, or therapeutic responses [19].

Liquid biopsy could be helpful in screening the effectiveness of treatment and predicting liver cancer progression. Nowadays, the diagnosis and identification of HCC rely on serum biomarkers (alpha-fetoprotein, or AFP) and imaging techniques such as ultrasound, computed tomography, and magnetic resonance imaging. However, small tumors or early-stage tumors are difficult to detect using imaging methods alone [20]. Thus, invasive procedures like liver biopsies are often required when there is ambiguity about the diagnosis through imaging techniques. In the future, liquid biopsy is anticipated to play a key role in HCC detection, monitoring, and treatment planning.

This study provides a comprehensive and up-to-date overview of recent advances in liquid biopsy technology and their clinical relevance in the diagnosis, prognosis, and therapeutic management of HCC. Additionally, it investigates the use of CTCs, cfDNA, cfRNA, and EVs as minimally invasive biomarkers in the diagnosis and prognosis of liver cancer. In addition, this review highlights current challenges, limitations, and future perspectives of liquid biopsy for HCC management.

## 2. Liquid Biopsy for Early Diagnosis and Management of HCC

In HCC, the balance between tumor suppressor genes and proto-oncogenes becomes disrupted, and many oncogenic signaling pathways become activated. The prognosis of HCC is typically poor since HCC typically develops in the context of a diseased or cirrhotic liver, and most patients get diagnosed with HCC at an advanced stage [21]. Early identification of HCC and early intervention are crucial because current curative treatments are most effective in early-stage HCC. However, current diagnostic techniques for HCC, such as biopsy, alpha-fetoprotein (AFP) testing, and radiological imaging, have several limitations [22].

Vaccination campaigns, antivirals against hepatitis C, and healthy lifestyle choices have reduced HCC incidence. Patients with early-stage HCC are considered to be eligible for potentially curative treatments such as radiofrequency ablation, surgical liver resection (LR), or liver transplantation (LT). Conversely, primary treatment for patients with advanced HCC (BCLC C) consist of systemic treatments, like tyrosine kinase inhibitors such as sorafenib or regorafenib and a combination of bevacizumab (anti-vascular endothelial growth factor monoclonal antibody) and atezolizumab (anti-programmed death-ligand 1-PD-L1-monoclonal antibody). Despite these treatments, median survival rates remain less than two years [23]. Therefore, early detection of HCC could be very beneficial for maximizing the survival of individuals [24]. During metastatic progression, early detection at an earlier stage could enable effective treatment [25].

In this regard, liquid biopsy is a highly promising technique for early detection of HCC, as well as for the limited liver tissue specimens for analysis. Methylation profiling of circulating tumor DNA has emerged as a promising monitoring method for early diagnosis of HCC in populations at risk, potentially revolutionizing the implementation of surveillance programs in the near future [26]. Circulating nucleic acid markers, such as cfDNA, microRNA (miRNA), long noncoding RNA (lncRNA), and mRNA, have been reported to be linked with malignant diseases [27]. Among the biomolecules released into bodily fluids, CTCs, circulating cell-free nucleic acids (cfDNA and cfRNAs), and circulating extracellular vesicles (exosomes) have great significance in diagnosis and prognosis [28]. Figure 1 outlines the diverse and evolving strategies employed in liquid biopsy.

## 3. Circulating Tumor Cells

Metastasis is a biologically complex and dynamic process that differs greatly among different cancer types and involves various stochastic events. Circulating tumor cells (CTCs) are cells that are released from primary tumors and metastatic sites into the bloodstream. While they are typically absent or rare in normal or non-malignant tissues, CTCs can be detected under certain pathological circumstances. They are sometimes called disseminated tumor cells (DTCs). From their initial intravasation into the bloodstream to the formation of clinically identifiable metastases, CTCs have the potential to significantly advance our knowledge of the stages involved in the metastatic cascade [29].

At tumor efferent arteries, CTCs are primarily epithelial, but under the effect of signaling pathways involving Smad2 and β-catenin, they can undergo transition to a mesenchymal phenotype [30], enhancing their migratory and invasive potential. The cells spread through portal venous and systemic circulations [31]. In the bloodstream, they undergo dynamic changes, including aggregation and disaggregation, as well as alterations in size and form. During extravasation, integrin-mediated interactions between CTCs and endothelial cells facilitate their exit. The invading CTCs can multiply and create visible metastatic lesions in a supportive environment, facilitating cancer cell survival and proliferation. In this case, the curative options become more limited. However, metastases become noticeable in only patients who have completed this entire process. The discovery of CTCs in patients with early-stage malignancies does not suggest an incurable cancer. This kind of aggressive cancer, identified by the presence of CTCs, needs the addition or augmentation of adjuvant systemic therapy for the eradication of any undetected micrometastases and improvement in the patient’s survival [32].

CTCs exhibit a number of molecular markers. Epithelial cell adhesion molecules (EpCAM) are the most often utilized molecular marker for CTCs because the majority of tumors are of epithelial origin. Ep-CAM-based CTC detection technologies are commonly used for malignancies such as breast and prostate cancer; however, EpCAM expression differs throughout cancer types. Ep-CAM-positive CTCs are also detected at a significant rate in other epithelial-derived cancer types, including hepatocellular cancer, colorectal cancers, and pancreatic cancers [33].

CTC enrichment techniques can be divided into two categories: positive and negative. The foundation of negative-enrichment techniques is the elimination of white blood cells by antibody-based depletion techniques, typically utilizing CD45 depletion cocktails or anti-CD45-coated magnetic beads [34,35,36]. Immunocytochemistry, quantitative real-time polymerase chain reaction (PCR), flow cytometry, and immunofluorescence in situ hybridization (iFISH) may combine multiple probes or antibodies to identify distinct CTC subpopulations and can subsequently detect enriched CTCs [34,37]. Positive-enrichment techniques use particular biological and physical characteristics of CTCs to differentiate them from blood cells that are not tumorous. Centrifugation and microfiltration are commonly used in physical enrichment techniques to identify CTCs based on their size and differential density, respectively. Immunoaffinity is the most used positive-enrichment technique, although it is unknown which biomarker best fits all CTC subpopulations [34,38]. Immunoaffinity is based on the presence of antibodies and detects CTCs with certain specific cell markers such as EpCAM or HER2. In brief, two basic strategies are used for the detection of CTCs, including biophysical-property-based detection techniques and antibody-based approaches. The size for filtration and density by centrifugation are common biophysical properties-based approaches. However, antibody-based approaches are based on the utilization of antibodies for the detection of cell surface markers present on the cell surface [39].

The mechanism behind the multifaceted, multi-step process of hepatoma cell invasion and metastases is yet unclear. The role of CTCs in the process of cancer metastases has been the subject of intense research in recent years [40]. Any cell expressing a liver or HCC-specific marker and circulating in the bloodstream may be regarded as a possible CTC. The liver-specific or HCC-specific proteome is a reasonable source of potential indicator for the identification of CTCs in HCC. Currently employed in clinical practice for pathological analysis and the characterization of HCC, GPC3 is a cell membrane-anchored protein [34,41].

The presence of GPC3-positive CTCs may indicate a dedifferentiated metastatic HCC, as GPC3 is predominantly found in moderately to weakly differentiated HCC tumor cells, which have a greater tendency for extrahepatic spread. Therefore, isolating GPC3-positive CTCs using immunomagnetic enrichment, followed by fluorescent active cell sorting and anti-cytokeratin (CK) (anti-CK7, anti-CK9, anti-CK20, and anti-CK8) staining, has been considered to be a significant approach for patients with HCC, an epithelial marker, and antibodies. Microvascular invasion [42], higher tumor recurrence rates, and reduced survival [43] are documented to be linked with increased numbers of CTCs [34]. EpCAM, typically absent in normal cells, is one of the most popular membrane-associated proteins for identifying CTCs in peripheral blood samples [44]. EpCAM-positive CTCs have been reported to be associated with aggressive features of HCC, such as vascular invasion, elevated serum AFP (≥400 ng/mL), and a more advanced BCLC stage [45,46].

Liver-derived CTCs from HCC patients have been used to analyze the expression of epithelial-to-mesenchymal (EMT) markers, including vimentin, twist, zinc finger E-box-binding 1 (ZEB1), zinc finger E-box-binding 2 (ZEB2), snail, slug, and E-cadherin. Notably, the co-expression of twist and vimentin correlates with tumor burden, macrovascular invasion, and advanced disease stages. The CanPatrol™ system, a combination of the RNA–in situ hybridization (RNA-ISH) technique and the positive-enrichment filter-based method, classifies CTCs into three phenotypic subgroups, including mesenchymal phenotype (vimentin, twist), mixed/hybrid phenotype, and epithelial phenotype (EpCAM, CK 8/18/19). Serum AFP is less accurate than total CTC count in differentiating HCC from non-malignant liver diseases [47]. Mesenchymal and hybrid CTCs are reported to be linked with poor prognoses, such as high blood AFP levels, advanced tumor stages, and microvascular invasion. Thus, EMT-based CTC categorization may serve as a predictive marker of early HCC recurrence, metastasis, and a reduced survival rate [34,48,49].

This classification is significant because mesenchymal CTCs are often detected in patients with advanced stages of the disease, providing information about treatment and prognosis. The need to identify distinct tumor cell subpopulations results from this, and while multiple approaches may be able to do so, it is crucial to standardize these disparate approaches [50]. Technological standardization is necessary before their biological specificity and clinical validity can be further examined. Another difficulty is that CTMs have a different role as prognostic and predictive biomarkers than CTCs [51].

The process of isolation and identification of CTCs lacks standardization and is a critical challenge for several reasons, including a lack of verified thresholds for marker expressions, variability in detection due to technical issues, and the unavailability of standardized protocols for clinical interpretations [52]. Recently, some studies have evaluated the clinical cutoff value for mesenchymal markers. A study in 2023 on colorectal cancer using cell-surface vimentin assay identified a count of CSV-CTCs ≥ 3 per 7.5 mL blood as an independent risk marker for worse prognosis [53]. In metastatic gastric cancer patients, ≥2.5 Twist-positive CTCs per 7.5 mL of blood was the worst prognostic marker [54]. However, the clinical applications of these phenotypes are limited due to the absence of standardized protocols and universally accepted thresholds.

Additionally, cell surface markers like EpCAM, CD133, CD44, CD90, or ICAM-1 may be expressed by circulating stem cells (CSCs). Among these, CD44 serves as an adhesion molecule that promotes tumor cell invasion as well as migration [34,55]. CD133-positive and CD44-positive cells (unlike CD133-positive CD44-negative cells) exhibit stem-cell-like features in HCC cells and tumor xenograft models, such as widespread proliferation, self-renewal ability, high tumors, chemoresistance, and the expression of genes linked to stem cells [56]. The elevated serum transaminases, serum AFP, and adverse clinical outcomes have been linked to CD133-positive, CD44-positive cCSC in HCC patients [57]. A summary of various investigations highlighting the diagnostic and prognostic significance of CTCs for various cancers is provided in Table 1. 

Although EpCAM-based techniques are usually implicated in the detection of CTCs, many CTCs found in patients with various cancers lack adequate epithelial features due to their epithelial-to-mesenchymal transition (EMT). During EMT, certain mesenchymal surface indicators, like N-cadherin, are elevated, while other epithelial markers, like EpCAM and E-cadherin, may be downregulated. Therefore, the cells acquire greater motility and invasiveness. Therefore, CTCs that have undergone EMT fail to be detected through the EpCAM-based CTC detection technique [58].
diagnostics-15-01655-t001_Table 1Table 1Summary of various studies, highlighting the results showing that CTCs are involved in the identification of different types of cancer.SampleNo of PatientsConclusion of InvestigationCancer TypeReferencesPeripheral blood
150
In cases with primary lung cancer, CTC is a helpful surrogate marker of distant metastases.
Lung Cancer
[59]Peripheral blood
174
CTC detection can help distinguish between benign and malignant lung nodules.
Lung cancer
[60]
Blood

63

It is possible to analyze cancer-related changes at the DNA and protein levels from CTCs. The changes seen in AR and EGFR imply that the approach might play a part in clinical judgment.

Metastatic prostate cancer
[61]Peripheral blood
179

Survivin expression increased, suggesting anti-apoptotic actions. Therefore, the prognosis and metastasis of HCC can be predicted by survivin-positive CTCs.

Hepatocellular carcinoma
[62]Peripheral blood
2662

Patients with CTC-positive HCC had a substantially worse outcome than those with CTC-negative HCC.

Hepatocellular carcinoma
[63]Peripheral blood
270

Proliferative CTC percentage (PCP) and CTC clusters may enhance the performance of the serum biomarker AFP and predict the recurrence of HCC.

Hepatocellular carcinoma
[64]Peripheral blood
29

A certain genomic profile was linked to the quantity of mesenchymal CTCs and CTC-WBC clusters in peripheral blood in individuals with hepatocellular cancer.

Hepatocellular carcinoma
[65]Peripheral blood
139

Increased postoperative CTC levels are linked to a worse prognosis for individuals with HCC, and surgical liver resection is linked to an increase in CTC counts.

Hepatocellular carcinoma
[66]
Blood

112

In Hepg2, Hep3B, and Huh7 cells, BCAT1 was markedly increased, and its knockdown promoted apoptosis while decreasing invasion, migration, and proliferation. BCAT1 may initiate the EMT process, as evidenced by a concurrent decrease in mesenchymal marker expression (vimentin and Twist) and an increase in epithelial marker expression (EpCAM and E-cadherin). In general, there was a strong correlation between CTCs and HCC traits.

Hepatocellular carcinoma
[67]Whole blood20In metastatic HCC, CTCs can be identified by EpCAM enrichment without being confused by a false-positive background from NMLD. Poor prognostic variables were linked to CTC detection.Hepatocellular carcinoma[46]Blood105In patients with early HCC undergoing surgery, ΔCTC, which is calculated based on the physical characteristics of the cells, is predictive of recurrence.Hepatocellular carcinoma[68]Peripheral blood52Finding positive CTCs can aid in forecasting the clinical course of HCC patients.Hepatocellular carcinoma[69]Peripheral blood73Higher tumor aggressiveness characteristics and a lower survival rate for patients with HCC are associated with a greater number of peripheral CTCs. In HCC, CTCs may develop into a unique prognostic biomarker.Hepatocellular carcinoma[70]Peripheral blood
152
With a positive predictive value and accuracy of over 90%, CTCs can be utilized as a predictive biomarker for oral cancer.
Oral squamous cell carcinoma
[71]Peripheral blood
186

Growing tumor size increases the likelihood of CTC identification, particularly in cases of clear cell renal cell carcinoma (ccRCC).

Renal Cell Carcinoma
[72]
Blood

47

LBD > 5 mm, subretinal fluid, orange pigment, sonographic hollowness, and the presence of multiple risk variables with *p* < 0.001 for all parameters were more common in the positive CTC group than in the negative CTC group.

Choroidal Melanocytic Lesions
[73]
Blood

95

One potential prognostic indicator of tumor spread in HNSCC patients is the identification of CTCs in these individuals.

Head and Neck Squamous Cell Carcinoma
[74]Peripheral blood
88

Encouraging outcomes when using FAST-based CTC detection for CRC prognosis and early diagnosis.

Colorectal cancer
[75]Peripheral blood
22

Viable CTCs can be used to identify CTC heterogeneity. It is possible to separate viable CTCs and cultivate them later in gastric cancer research.

Gastric cancer
[76]

## 4. Circulating Cell-Free Nucleic Acids as Non-Invasive Biomarkers in Cancer and Disease Monitoring

Cell-free nucleic acids comprise a variety of DNA and RNA molecules found in extracellular fluids [77]. Apart from active transport pathways, these nucleic acids are produced by cellular pathways such as necrosis or apoptosis. After their release, nucleic acids form complexes with proteins or are enclosed in extracellular vesicles or microvesicles (exosomes). The formation of complexes or exosomes provides stability to cell-free nucleic acids in bodily fluids. This stability makes them appropriate biomarker candidates for detection using techniques like qPCR and sequencing. Cell-free nucleic acids (cf-DNA and cf-RNAs) have been shown to be useful biomarkers for the diagnosis and prognosis of diabetes, cancer, and neurological or cardiovascular disorders [78].

Because of its relative fragility, cfRNA is typically discovered in conjunction with proteins, proteolipid complexes, and EVs, whereas cfDNA can be found in circulation as long pieces encapsulated in EVs or small fragments linked with nucleosomes [79]. One extremely promising liquid biopsy method for learning more about liver cancers is the study of circulating nucleic acids. cfDNA and cfRNA are excellent choices for tumor molecular profiling in addition to their use in risk prediction, early identification, and treatment response monitoring. Their capacity to reflect tumor heterogeneity, in contrast to tumor biopsy, makes them an effective tool for detecting chromosomal abnormalities, point mutations, and aberrant methylation that confer drug resistance and direct molecular target therapy [80]. Cell-free nucleic acids are utilized in both cancer diagnostics and treatments [78] (Figure 2). cfDNA and cfRNA have been reported to be involved in the detection and diagnosis of several cancers (Table 2).

### 4.1. Circulating Tumor DNA (ctDNA): Origin, Characteristics, and Clinical Potential

Any non-encapsulated DNA found in the bloodstream that comes from different cell types is referred to as circulating cell-free DNA (cfDNA). cfDNA is non-cell-associated DNA and is found freely circulating in blood and other body fluids. cfDNA is released by normal or tumor cells through apoptosis, necrosis, or secretion. cfDNA can be either ctDNA or non-tumor DNA. Structurally, cfDNA can be single-stranded or double-stranded [108]. ctDNA is a subset of cfDNA, and it consists of tumor-derived DNA fragments. ctDNA is found freely circulating in both plasma and serum. ctDNA carries cancer-associated characteristics such as DNA methylation, some viral sequences, and single-nucleotide mutations, possibly because they originate from the tumor tissues [109]. ctDNA has drawn more attention due to its ability to identify early cancer metastases using sensitive, new laboratory techniques that high-resolution imaging cannot [110].

Solid tumors can leak DNA fragments into the bloodstream through both active and passive methods. The latter appears to be more prevalent and is fueled in part by apoptosis and cell necrosis. This process happens during tumor development and provides minimally invasive access to critical genomic (point mutations or copy number variations, or CNV) and epigenetic (e.g., changes in DNA methylation) information. The usefulness of ctDNA as a polyvalent biomarker in cancer has been demonstrated by numerous research studies [111]. The presence of genetic or epigenetic modifications unique to cancer is frequently used to distinguish ctDNA from non-neoplastic cfDNA. Therefore, ctDNA is highly focused for biomarker discovery and clinical applications in various cancers, including HCC [112].

A number of studies have shown that liquid biopsy can identify specific circulating DNA alterations that are directly linked to a particular tumor. This is made possible by the fact that ctDNA has a short half-life, which enables instant correlation with tumor cell status and, consequently, provides the opportunity for ongoing dynamic surveillance [113]. Compared to a traditional tissue biopsy, which has the inherent spatial limitation in sampling because of tumor tissue heterogeneity, ctDNA analysis can yield more thorough information. Up to 3.3% of tumor DNA, or 3 × 1010 tumor cells, are thought to enter the bloodstream every day from 100 g of tumor tissue [114].

Mutated tumor cells release ctDNA into the bloodstream and the genetic profile of the original tumor. The detection of this liberated ctDNA in the bloodstream offers a more sensitive and specific, noninvasive method for cancer diagnosis and prognosis and the monitoring of therapy responses. Additionally, its identification in the blood aids in tumor diagnosis and prognosis as well as targeted therapy [115]. ctDNA levels are significantly increased in patients compared with healthy people and are considered to be a possible biomarker. According to certain research, it is associated with tumor volume, which reduces overall survival, and its value is elevated in colorectal, breast, ovarian, lung, and liver cancer. Additionally, ctDNA may provide valuable insight into tumor burden after therapy or surgery and treatment monitoring and may identify early cancer. Noninvasive collection of ctDNA, which allows continuous patient monitoring, can offer a significant advantage over traditional needle biopsies [116].

For early cancer detection, treatment monitoring, and tumor burden prediction, ctDNA can be very important. ctDNA carries tumor-specific genetic and epigenetic alterations [117]. The length of the cfDNA ranges from 18 bp to 10,000 bp and has a half-life of less than two hours in the bloodstream [118]. Common genomic alterations detectable in ctDNA include single-nucleotide variations (mainly somatic variations) in the genes TP53, KRAS, and CCND1; copy number variations involving genes such as CDK6, EFGR, MYC, and BRAF; DNA methylation changes in genes such as RASSF1A, SEPT9, KMT2C, and CCNA2; homozygous mutations (e.g., CDKN2A and AXIN1); and large chromosomal rearrangements, which are especially implicated in HCC [117].

In HCC, specific copy number gains (chromosomes 1, 7, 8, and 20) and losses (chromosomes 4, 8, 13, and 17) have been linked with tumor progression. Liquid biopsy samples can yield cfDNA amounts of up to 100 ng/mL in healthy people and of up to 1000 ng/mL in cancer patients [119]. Recent studies have documented that mutations in ctDNA (e.g., TERT, TP53, and CTNNB1) are detectable by droplet digital PCR (ddPCR) in HCC patients. However, tumor heterogeneity and tissue biopsy sampling errors may lead to inconsistencies. For example, TERT promoter mutations (C228T and C250T) have been reported in plasma cfDNA of 218 patients with early- and late-stage HCC by ddPCR [120].

ssDNA primarily originates from many sources, including tumor cells [121], a variety of normal cell types, including immune cells (such as T cells, B cells, and macrophages) [122], endothelial cells [123], and others due to natural cell turnover, apoptosis, or immune response [124], as well as extracellular vesicles, including exosomes and microvesicles, secreted by both cancer and normal cells [125]. ssDNA describes a physical structure, and it does not refer to the cellular origin or diagnostic relevance. So the term ssDNA should be used carefully.

### 4.2. Circulating and Tumor-Derived RNAs as Emerging Biomarkers in Cancer and HCC

Circulating RNAs refer to RNA that is found in body fluids. Circulating tumor RNAs are RNA fragments released by tumor cells into the bloodstream. As unstable molecules, circulating-free RNAs are rapidly broken down by ribonucleases. It was documented that 99% of naked RNA was broken down after 15 s of incubation. However, endogenous circulating RNAs are shielded from nuclease activity by a number of mechanisms, including encapsulation within extracellular vesicles (EVs) or the formation of ribonucleoprotein complexes with RNA-binding proteins like nucleophosmin, high-density lipoprotein, or Argonaute 1 and 2 [126]. However, because RNA expression is complex and ephemeral, this paradigm has only lately come of age. Circulating RNAs are remarkably stable in plasma, possibly due to their interaction with exosomes or subcellular structures [127].

Messenger RNA and noncoding microRNA (miRNA) are examples of circulating RNAs [77]. Numerous biological processes, such as tumor growth, epithelial–mesenchymal transition, and cell proliferation, have been shown to depend critically on circular RNAs. Perhaps as a result of their interaction with subcellular particles or exosome packing, they are found in plasma in a very stable form. Short fragments make up the majority of the mRNA present in plasma. Numerous tumor-derived cell-free mRNA products, such as CCND1, Her2/neu, and TERT, have been discovered [127]. ctRNA encompasses both protein-coding messenger RNA (mRNA) and various non-coding RNAs such as circular RNA (circRNA), long non-coding RNA (lncRNA), and microRNA (miRNA), which differ in their transcript size and structure. ctRNAs are recorded as promising biomarkers in cancer. One of the main benefits of ctRNA analysis is the ability to profile RNA expression, which in turn offers important information about the overexpression of transcripts specific to cancer. This is because ctRNAs primarily function as signaling molecules in cell-to-cell communication in the extracellular matrix. Hence, ctRNAs are promising biomarkers for tracking tumor progression, assessing response to treatment, and enabling early cancer detection [128].

One helpful source of cancer biomarkers may be circulating mRNA. Some patients with melanoma may have tyrosinase mRNA in their blood [129]; patients with breast cancer may have telomerase mRNAs [130]; patients with thyroid cancer may have CK19, mammaglobin, 5T4, and circulating thyroid-stimulating hormone receptor (TSHR) mRNAs [131]; patients with lung cancer may have 5T4 mRNA [132]; and patients with colorectal cancer may have CEA and CK19 mRNAs [133]. One study offered a novel biomarker, plasma PCTK1 mRNA, and illustrated a method for identifying circulating RNA markers [27].

Alpha-fetoprotein (AFP) and other blood biomarkers, as well as imaging technologies like computed tomography and ultrasound, are key components in the clinical diagnosis of HCC. However, the low sensitivity of the aforementioned conventional approaches makes it difficult to identify HCC at an early stage [134]. HCC is usually discovered when it has progressed, and there are few viable therapeutic options. Finding reliable biomarkers for the early detection of HCC is crucial. There is growing evidence that ncRNAs, such as circular RNAs (circRNAs), long non-coding RNAs (lncRNAs), and microRNAs (miRNAs), may be utilized in the diagnosis of HCC [135]. According to a meta-analysis, miRNAs have sensitivity, specificity, and AUROC values above 80%, making them diagnostically accurate for HCC on par with conventional biomarkers [136]. Circular RNA (circRNAs) is the most effective, according to a network meta-analysis, with miRNAs and lncRNAs coming in second and third, respectively [137]. While early diagnosis can be very crucial for improving HCC patients, a reliable, high-throughput screening approach needs to be established [138].

In three cohorts, to find prognostic markers, the relationships between circulating microRNAs (miRNAs) and liver-damage markers, clinicopathological features, and survival outcomes were examined. In all three cohorts, there is a difference in the expression of twelve miRNAs between NH and HCC individuals. In patients with CHB, cirrhosis, and HCC, five elevated miRNAs (miR-122-5p, miR-125b-5p, miR-885-5p, miR-100-5p, and miR-148a-3p) may be biomarkers for CHB infection, whereas miR-34a-5p may be a biomarker for cirrhosis. Interestingly, four miRNAs—miR-1972, miR-193a-5p, miR-214-3p, and miR-365a-3p—can differentiate people with HCC from those without. Potential prognostic indicators for overall survival include six miRNAs [139].

VEGF expression level (isoform 165) was found to be associated with the likelihood of tumor recurrence in a comparison of 50 HCC patients and 50 controls [140]. Two more research studies looked into the amount of circulating messenger RNA (mRNA) that codes for AFP. The first one, which involved 38 HCC patients undergoing partial resection, demonstrated a correlation between AFP mRNA detection and a lower disease-free survival and extrahepatic recurrence [141]. AFP and hTERT mRNA levels were evaluated in the second one, but no predictive effect was found [142].

### 4.3. Detection and Analysis of Cell-Free Nucleic Acids

The analysis and detection of cell-free nucleic acids are important methods for liquid biopsy across various clinical fields. Until now, three main methods have been employed for the absolute quantification of cell-free nucleic acids: NGS with subsequent bioinformatic analysis, digital droplet PCR (ddPCR), and real-time quantitative PCR (RT-qPCR). These methods offer high sensitivity and specificity and enable the detection of low-abundance nucleic acid fragments. Additionally, the MassARRAY system, a matrix-assisted laser desorption ionization time-of-flight mass spectrometry (MALDI-TOF MS) tool, has been effectively applied for the identification of cell free fetal DNA (cffDNA) with significant accuracy [143,144].

## 5. Extracellular Vesicles or Exosomes: Implications for Cancer Progression and Biomarker Discovery

Exosomes are subtypes of extracellular vesicles (EVs), and they carry numerous biomolecules, such as DNA and RNA, and are protected from enzymatic breakdown by the presence of a lipid bilayer. The unique characteristics of circulating exosomes protect RNAs from degradation, allowing stable transport through the plasma. Exosomes may be actively released into the bloodstream from apoptotic cells, although their exact biological nature and functions are still not fully understood [111]. EVs can be captured utilizing cutting-edge methods like magnetic beads, aptamers, and microfluidics, which enable high-throughput multi-omics analysis and make it easier to employ them for biomarker identification [145].

All live cells secrete EVs and are present in every bodily fluid. Their size, composition, and biogenesis vary greatly, but their cell-of-origin-particular cargo loading always reflects that of their parent cells. EVs are being considered more and more as useful carriers of cancer biomarkers in liquid biopsy samples since a number of studies have shown that EV-associated proteins, nucleic acids, lipids, and metabolites can indicate malignant phenotypes in cancer patients. EVs not only play a role in pathological processes like cancer, but also they contribute to physiological processes. Consequently, EV-based liquid biopsy can be used to learn more about the tumor and its growth [146].

EVs can be divided into groups based on their size or cellular origin. While vesicles that are directly released from the plasma membrane are referred to as microvesicles, EVs originating from the intracellular endosomal system are traditionally referred to as exosomes. EVs up to 200 nm in size are now simply known as “small EVs” (sEVs), whereas vesicles larger than 200 nm are known as “large EVs” (lEVs), as this distinction was not always evident [125]. Another kind of nanoparticle, known as exomeres, has just been described; these particles are smaller than 50 nm. The purpose of these non-membranous particles is currently unknown [147].

EVs are also secreted by tumor cells, and EVs produced from tumors have been linked to angiogenesis, metastasis, and resistance to treatment [148]. In the tumor microenvironment, they can impact nearby cells, as well as distant cells and organs, where they aid in the development of pre-metastatic niches [149]. It has been discovered that EV biogenesis involves certain proteins that are overexpressed in malignancies. It has been demonstrated that EVs generated by tumor cells induce extracellular matrix (ECM) remodeling, suppress immune system responses, and encourage angiogenesis, all of which contribute to the development of a tumor-supportive microenvironment [146]. Furthermore, tumor-derived EVs have the potential to transmit medication resistance, making them attractive targets for upcoming therapeutic strategies [150]. The molecular cargo of tumor-derived EVs, which includes metabolites and tumor-specific proteins, microRNAs, or mRNAs, mediates these harmful consequences [150,151].

EV samples were separated based on their cellular origin by the proteomic examination of tumor-derived EVs with simultaneous hierarchical clustering, which has demonstrated that tumor-derived EVs reflect the molecular makeup of the secreting cancer cells in several investigations. Comparable profiles of RNA and DNA have been detected in EVs released from various cancerous types, describing their significance in tumor biology [146]. The expression of vesicular copine-3 (CPNE3) in CRC patients’ plasma samples has been found to be favorably connected with both overall survival and the protein signal from the matching tumor tissue [152]. EV-DNA generated from plasma was reported to be more sensitive than cfDNA in detecting Kirsten rat sarcoma (KRAS) mutations in PDAC patients [153]. Circulating EV-based biomarkers are also thought to effectively represent the residual tumor cells in cancer patients’ blood following treatment because of their strong diagnostic relevance [146].

Up to 20% of HCC cases are discovered in non-cirrhotic individuals, while most HCC diagnoses occur in patients with underlying cirrhosis. In order for hepatocytes, stellate cells, and different immune cells (Kupffer cells, T and B cells, and natural killer—NK—cells) to carry out vital tasks and preserve a homeostatic state, EVs are essential for mediating a variety of signals in a healthy liver. According to earlier research, EVs contribute to the emergence of these predisposing hepatic disorders and, in turn, to the development of HCC. By controlling the microenvironment and several signaling pathways in both the malignancy and the surrounding normal cells, EVs regulate the etiology and progression of HCC [20].

Multiple studies suggest that EVs may play a direct role in the development of HCC through various molecular pathways [20,154]. EVs released by HCC cells transfer long intergenic non-protein coding RNA, regulator of reprogramming (linc-ROR), which stimulates normal hepatocyte growth and inhibits apoptosis. Hepatocytes cocultured with HCC-derived EVs for over 30 days showed a significant increase in the expression of stem cell-related proteins, including OCT4, NANOG, SRY-box 2 (SOX2), P53, and CD133 [20].

EV protein is reported to be a viable biomarker for identifying an unusual intrahepatic lesion that occurs between iCCA and HCC. SMAD3, one of the molecules promoting HCC metastasis, also has diagnostic power for HCC (AUC of 0.70 for differentiating HCC from healthy controls and benign hepatoma) [155]. miRNAs encapsulated in EVs have been demonstrated as promising biomarkers for HCC diagnosis. In their analysis of the miRNA profiles of EVs from patients with cirrhosis and HCC, Wang et al. discovered that some elevated miRNAs (miR-122, miR-148a, and miR-1246) were more effective than AFP at differentiating between cirrhosis and HCC. The final panel, which included AFP, miR-122, and miR-148a, produced an AUC of 0.93 [154]. MiR-155, lncRNA-H19, and circRNA-100338 from HCC-derived EVs are linked to angiogenesis in cell line studies [156,157,158]. However, by downregulating ERG and LPIN1, respectively, miR-200b-3p and miR-451a inhibit angiogenesis [159]. A signaling protein called vascular endothelial growth factor (VEGF) directly stimulates the proliferation of hepatocytes, cancer cells, and epithelial cells, resulting in aberrant vascular architecture in HCC [160].

By transferring miR-1228-3p to HCC cells, EVs produced from cancer-associated fibroblasts (CAFs) increased patient resistance to sorafenib by enhancing HCC invasion, migration, and proliferation through the activation of the PLAC8-mediated PI3K/AKT pathway [161]. In these EVs made from the urine of patients with HCC, the expression of the glycoproteins LG3BP, PIGR, and KNG1 was considerably increased, whereas ASPP2 expression was significantly downregulated. Additionally, it was demonstrated that aberrant EV membrane protein glycosylation has the potential to be a useful noninvasive diagnostic for the diagnosis of HCC [162].

For miRNA-level sequencing, Lin and colleagues separated EVs from HCC and nearby liver tissues. EVs extracted from the serum of both healthy participants and HCC patients were also used to sequence the miRNA levels. The only differently expressed miRNA found in EVs from HCC tissue and patient plasma, according to an analysis of sequencing data and experimental validation, was hsa-miR-483-5p. Subsequent investigation revealed that miR-483-5p was highly abundant in HCC EVs and that it facilitated the growth of HCC cells by binding to CDK15 and suppressing CDK15 production, eventually driving the malignant development of HCC. Hsa-miR-483-5p was shown in this study to be a possible biomarker for HCC diagnosis [163]. Table 3 summarizes the conclusions of investigations utilizing EVs for the detection of various cancers.

Exosome isolation methods include a variety of methods or their combinations. A commonly used method is ultracentrifugation, which applies sequential centrifugation speeds (300× *g*, 2000× *g*, 10,000× *g*, and 100,000× *g* for varying durations) to progressively remove cells, debris, and larger vesicles. Other methods include density-gradient filtration, which separates on the basis of buoyant density, and size exclusion chromatography, offering reproducibility and maintaining integrity. Additionally, membrane filtration techniques purify vesicles on the basis of size. Advanced techniques such as aptamer-based microfluidics and immunoaffinity capture purify vesicles on the basis of the presence of markers unique to exosomes. Magnetic bead-based isolation allows selective isolation of exosomes with minimal handling [125].

The lack of standardized protocols and university-accepted isolation protocols is a major challenge faced by researchers in clinical isolation of EVs, including exosomes, which limits their clinical implications. Current methods of isolation vary in terms of yield, purity, and reproducibility, leading to inconsistencies in results across various clinical and research settings. More recent methods like size-based isolation and immunoaffinity capture have drawbacks of their own, including the possibility of sample contamination and high expenses. While automated and high-throughput technologies simplify the isolation process and guarantee consistency and scalability, nanotechnology-based approaches provide novel alternatives by utilizing nanoscale materials to improve the specificity and efficiency of exosome capture. These sophisticated technologies are especially useful in clinical diagnostics and extensive research because they lower human error, increase reproducibility, and process massive sample volumes. Notwithstanding these developments, there are still issues with standardizing procedures for various techniques and guaranteeing the stability and biocompatibility of materials used in isolation procedures [171].

## 6. Tumor Educated Platelets and Their Role in Tumor Biology and Liquid Biopsy Applications

The second most prevalent cellular component in peripheral blood is blood platelets. Platelets are circulating enucleated cell fragments that come from bone marrow megakaryocytes. Tumor growth, invasion, and distant metastasis are all impacted by platelets’ interactions with tumor cells [172,173]. The sequestration of tumor-associated molecules by platelets facilitates their interaction with tumor cells. In fact, circulating mRNA can be directly ingested by platelets [174]. Furthermore, external signals such as lipopolysaccharide and platelet surface receptors can induce certain splice variants of pre-mRNAs in circulating platelets, forming distinct mRNA profiles that may be beneficial for cancer diagnoses [175]. Platelets have been demonstrated to play a significant biological role at many stages of malignant disease, including angiogenesis, cell proliferation, cell invasiveness, and metastasis [176].

Biomarkers present on the platelet surface and/or within the platelets are considered to have a crucial role in the development and metastasis of cancer [177]. At the same time, interactions between cancer cells and platelets can change the protein and RNA profiles of platelets [176,178]. Thus, platelets serve as crucial reservoirs of RNA species (microRNA, circular RNA, long noncoding RNA, messenger RNA, and mitochondrial RNA) and protein biomarkers, which have potential for early cancer detection, disease monitoring, and therapeutic responses. Additionally, a number of encouraging investigations demonstrated the aberrant clinical characteristics (such as platelet count and mean platelet volume) of early-stage cancer patients’ platelets [179,180].

After activation, many bioactive proteins, including growth factors, chemokines, and proteases, are released by platelets into the tumor environment [181]. Studies conducted on tumor-bearing animals in vivo revealed elevated levels of tumor-derived factors, including TGF-β, MCP-1, RANK, TIMP-1, and TSP-1, in platelets [182,183].

Following tumor removal, the expression level of TSP-1 dropped and showed a strong correlation with tumor progression [181,183]. In order to identify tumors that are clinically undetectable (less than 1 mm^3^) in mice, the overexpression of TSP-1 and PF-4 inside platelets was examined [183,184].

The potential of TGF-β, NF-κβ, VEGF, AKT, and PI3K as RNA-based biomarkers in tumor-educated platelets for the early identification of HCC was examined in a study. The transcriptional analysis’s findings showed that TGF-β, NF-κβ, and VEGF were significantly overexpressed compared to the control, by 2.48, 2.35, and 2.78 fold, respectively. In contrast, AKT and PI3K showed 0.6- and 0.65-fold reductions in expression, respectively, when compared to controls [185]. Tumor-educated platelet (TEP) microRNA (miRNA) expression and its possible diagnostic value in HCC were investigated in a bioinformatics-based study. Patients with HCC showed differential expression of TEP miRNAs, including miR-495-3p and miR-1293, which may contribute to the pathogenesis of HCC [186]. Higher VWF and enhanced platelet aggregation are linked to HCC in cirrhosis patients [187].

To distinguish early HCC from advanced-stage cirrhotic nodules, researchers evaluated the diagnostic validity of a specific platelet mRNA expression profile. IFITM3 and SERPIND1 displayed a 2.24-fold change, while RhoA, CTNNB1, and SPINK1 showed a substantial 3.3-, 3.2-, and 3.18-fold overexpression, respectively, in HCC patients compared to cirrhosis patients. Surprisingly, there was also a noticeable difference in the expression of CD41+ platelets between the groups with cirrhosis and HCC [188].

## 7. An Overview of Current Management of HCC

Hepatic I/R damage may be negatively affected by constitutive CYP2E1 in the liver [189]. The treatment options for HCC are mainly determined by the stage of HCC, and early, intermediate, and advanced stages require different therapeutic options. Potential treatment options for early-stage HCC include local ablation, transplantation, and surgical resection. Liver transplantation, which treats both HCC and underlying cirrhosis and delivers good survival rates with low recurrence risk, is limited by organ scarcity. Intra-arterial therapies, including transarterial embolization, transarterial chemoembolization, and transarterial radioembolization, can be used as bridging therapies before transplantation or as first-line treatments for patients with intermediate-stage HCC [190].

Among all therapeutic strategies for HCC, liver transplantation and surgical excision are the most effective options. Additional treatment possibilities include radiation therapy, stereotactic radiotherapy, systemic chemotherapy, transarterial chemoembolization, cryoablation, radiofrequency ablation, microwave ablation, percutaneous ethanol injection, and molecularly targeted therapies. Tumor location and size, extrahepatic dissemination, and underlying liver function all play a role in how HCC is managed [191].

Stereotactic body radiation therapy (SBRT) has become a bridging treatment for hepatocellular carcinoma patients waiting for liver transplantation (LT) in order to guarantee that patients retain their eligibility and priority position in accordance with Milan criteria [192]. For HCC patients on the waiting list, SBRT was a safe and efficient bridging treatment. Compared to other bridging therapies, SBRT resulted in higher rates of pathological complete response (pCR) in explant histology, improved one-year tumor control, and a decreased risk of patient dropout. As a safe and efficient substitute for TACE and HIFU in the existing bridging therapy regimen, SBRT should be added for individuals with HCC [193].

A key component of cancer treatment in recent years has been anti-cancer immunotherapies, which are therapeutic interventions intended to alter the immune system in order to identify and eradicate cancer. Immunotherapy has shown promise in improving survival and providing long-lasting cancer control in several HCC patient groups while lowering negative side effects. These findings document a major advancement in increasing treatment outcomes. Immuno-checkpoint inhibitors (ICIs) have been shown in clinical trials to extend survival in a subgroup of patients with HCC, especially when used in conjunction with tyrosine kinase inhibitors and anti-angiogenic agents. This offers a substitute for patients who do not respond to first-line therapy. Systemic treatments comprising immune checkpoint inhibitors and targeted medicines are frequently used in advanced-stage HCC. Treatment options for patients with chronic HCC have significantly increased since the advent of immunotherapy-based combination therapy [194].

However, HCC treatment is still difficult, and liquid biopsy gives a valuable insight into treatment-associated genetic changes of tumors, epigenetic information, tumor microenvironments, and resistance mechanisms using molecular foundation, circulating tumor cells, circulating DNAs, epigenetic profiles, and omics investigations [10,12]. For instance, resistance to sorafenib and other kinase inhibitors can be monitored through miRNA signatures in EVs [195,196]. Additionally, the molecular ctDNA might be a biomarker for anticipating drug resistance to sorafenib and other kinase inhibitors. EMT-mediated resistance to sorafenib in patients with advanced HCC can be triggered by changes in DNA methylation in cell lines. The challenge of re-acquiring and analyzing biopsy specimens of patients with advanced HCC is avoided with this non-invasive approach of obtaining genetic drug resistance information [108]. PD-L1 on large extracellular vesicles is a prognostic biomarker for therapeutic response in non-small cell lung cancer patients with PD-L1-low and -negative tissue. Further, the efficacy of immunotherapy might be evaluated using the evaluation of alterations in circulating immune-linked biomarkers, including PD-L1-positive EVs or T-cell-associated biomarkers. Patients who had elevated blood levels of PD-L1 + lEVs at baseline responded better to immunotherapy and had longer survival times [197]. Therefore, the combination or integration of liquid biopsy with current therapeutic approaches for the treatment of HCC could increase precision medicine in HCC by supporting more timely and individualized therapeutic decisions.

Several non-invasive biomarkers and composite scoring systems have been evaluated for the validation of early HCC diagnosis and risk stratification. HCC in HBV patients has been predicted using a number of risk scores. Only a small percentage of scores incorporate liver stiffness (LSM-HCC and LSPS), while the majority include basic clinical and biological information (age, male sex, platelet count, cirrhosis, or viral load). The only externally validated scores are the PAGE-B score (age, sex, and platelet count) and the modified PAGE-B score (albumin, age, sex, and platelet count) [198]. The only clinical and biochemical factors that substantially separated women with cancer from those without cancer were AST and non-invasive liver fibrosis scores (AARPRI, APRI, FIB-4, mFIB4) [199]. Patients who developed HCC showed higher liver transaminases and fibrosis scores at time 0 than those who did not [200].

## 8. Liquid Biopsy as a Contemporary Tool to Tissue Biopsy

In more than 90% of instances, HCC is diagnosed without a liver biopsy. Although Alpha-fetoprotein (AFP) and ultrasound screening at 6-month intervals are recommended as standard, they may not provide sufficient diagnostic accuracy for patients awaiting orthotopic liver transplantation. Because of their greater sensitivity and specificity, AFP, AFP-L3%, and/or des-gamma-carboxy prothrombin are detected in conjunction with triple-phase computed tomography and/or magnetic resonance imaging [191]. Currently, tissue biopsies are the gold standard method for confirming a cancer diagnosis. Liquid biopsy has emerged as a promising substitute for cases where tissue sampling is very challenging, especially for primary-stage tumors or determining metastasis progression. Additionally, it might reduce the side effects, bleeding, and infections that come with invasive tissue biopsies [201].

Unlike tissue biopsies, liquid biopsy is a non-invasive method for the detection of tumor recurrence and real-time monitoring of tumor dynamics and treatment response through the analysis of specific biomarkers in body fluids [202]. However, a tissue biopsy may still be necessary for confirmation, because a tumor might not be detected by liquid biopsy in some cases [12].

Liquid biopsies have a number of benefits over conventional tumor tissue analysis, such as quicker turnaround times, less invasive techniques, a more thorough evaluation of the mutational profile, and the potential for serial measurements that can more accurately depict tumor development [203]. As a result, liquid biopsy is a useful adjunct to tissue samples, and its evidence base for its many uses in cancer treatment is expanding quickly. A comparative summary of liquid biopsy and tissue biopsy is provided in Table 4.

## 9. Challenges and Limitations of Liquid Biopsy for Detection of HCC

Solid biopsies are currently accepted as a standard norm for clinical cancer diagnosis and its future therapeutic management. They offer data on tumor histology, molecular profiles for prognostic and predictive signatures with the best cost-effectiveness ratio, and standard biomarkers for subtyping and treatment planning. However, because of tumor heterogeneity, the molecular and genetic information about the tumor is restricted to the biopsy area, which could skew the interpretation. Furthermore, solid biopsies necessitate an invasive technique that could be hazardous and painful for the patient, and they are incompatible with longitudinal monitoring [205]. Comparatively, liquid biopsy is a unique method for the early screening, diagnosis, and therapy monitoring of liver cancer, especially when standard biopsies are not possible [206].

There are certain challenges associated with the use of liquid biopsy. Standardized procedures and quality control measures are absent from liquid biopsies. During sample collection, handling, processing, and storage, all liquid biopsy biomarkers are susceptible to degradation, which may cause contents to be lost or altered. Accurate analysis depends on these components’ stability. The liquid biopsy biomarker analysis may potentially be tainted by other analytes, including protein aggregates, lipoproteins, leukocytes, and cell-free DNA. Improving the specificity of isolation techniques is essential to lowering contamination and boosting the dependability liquid biopsy tests [207].

There are various current clinical trials (NCT05431621, NCT05575622, NCT02973204, ACTRN12615000381583, and ACTRN12617001566325) around the world that aim to evaluate the clinical value of CTCs and ctDNA as diagnostic and prognostic tools for HCC, as well as to guide clinical decisions. However, liquid biopsy biomarkers encounter additional hurdles in current clinical practice. Given the rarity of liquid biopsy biomarkers in blood, one important challenge is a lack of uniformity in techniques for sample preservation, enrichment, and detection [208].

Finding the ideal balance between sensitivity and specificity is an additional crucial problem. While high specificity makes sure that people without the ailment are not mistakenly recognized, high sensitivity is necessary to detect all cases of a disease or mutation [209]. Other particular difficulties are in isolating and standardizing EVs and miRNAs [210]. Although these molecules and vesicles provide important information about the genetic and molecular makeup of the tumor, their detection and quantification call for precise and standardized techniques [208]. Additional difficulties include the possibility of nontumor DNA contamination, the dynamic character of cancers that necessitates prompt capture for precise diagnosis, and accessibility and cost concerns [211].

The difficulty of reproducing a favorable milieu that promotes CTC survival and multiplication presents another problem in CTC investigation. Thankfully, new models that closely resemble the in vivo milieu, such as patient-derived organoids (PDOs) and patient-derived xenografts (PDX), have emerged, opening up new research opportunities for CTCs [212]. There are several opportunities for therapeutic use because CTC-derived PDX and PDO models allow unusual CTCs to grow in vitro and make drug sensitivity testing easier without requiring a biopsy. Although these models have been thoroughly studied in other cancer types, there is currently little use of them in liver cancer research [212,213].

The requirement to separate, purify, and identify the markers used in the monitoring procedure is a major barrier to the practical application of liquid biopsy. Therefore, in order to improve the accuracy of liquid biopsy in the future, it is essential to give priority to the development of new detection technologies and analysis platforms, together with the implementation of standardized operating protocols and unified data analysis. Additionally, researchers could attempt to connect it with the rapidly developing artificial intelligence, which could be a more successful detection method [214]. Regarding clinical use, it is crucial to remember that a liquid biopsy cannot adequately capture the complexity of a disease and can only reveal details about particular molecules or biomarkers [215].

Currently, in vitro sequencing and analysis—such as identifying multiple somatic mutations and including DNA methylation or fragmentation patterns—are the primary focus of research on increasing the sensitivity of ctDNA detection [216]. The low concentration of ctDNA in the drawn blood samples, which restricts sensitivity, is an inherent problem with all of these techniques. The sensitivity of the detection can be increased by increasing the volume of the blood sample. However, it is not practical for patients who are weak or sick. Furthermore, some have suggested techniques that accelerate tumor DNA loss or are more similar to tumor sampling. These techniques are also confined to particular main tumors, necessitate invasive surgery, are expensive, and require prior knowledge of the tumor location [217].

Lastly, compared to tissue molecular diagnostics, liquid biopsies are typically less costly and have faster turnaround times. Tumors may release small amounts of components, which makes detection in peripheral blood more difficult. This is a significant drawback of liquid biopsies. Additionally, different tumor sites may shed different components of the tumor, making it more difficult to assess tumor heterogeneity. Numerous assays have a wealth of evidence supporting their analytical and clinical validity across a broad range of tumor entities and indications; nevertheless, their clinical value in the majority of scenarios has not yet been confirmed [218].

For liquid biopsy to become a commonplace part of clinical practice, a number of obstacles need to be overcome, such as guaranteeing cost-effectiveness, assessing clinical value, putting laws in place, and developing a testing laboratory infrastructure. The integration of liquid biopsy into clinical practice has been impeded by the paucity of cost-effectiveness validation studies and the lack of pre-analytical and analytical standards brought on by the diversity of liquid biopsy analytes. The choice of blood collection tubes, sample transit timing, plasma or serum use, storage condition guidelines, cfDNA purification/quantification techniques, and sample preparation processes are all pre-analytical considerations [219].

## 10. Next-Generation Sequencing and Liquid Biopsy

Multiple gene mutations can be found thanks to NGS, which is based on the vast parallel sequencing of millions of distinct DNA molecules. Each read is sequenced thousands of times, guaranteeing a high level of sensitivity, by focusing the coverage on clinically important targets using targeted gene panels. After a careful validation process including blood collection, cfDNA extraction, library preparation, sequencing, and variant calling, NGS can represent a new gold standard technique for cfDNA [220].

NGS has high throughput and can screen unknown variations. MAF < 1% can currently be detected by NGS. Furthermore, numerous strategies like unique molecular IDs or unique barcodes can help to boost the sensitivity and reduce the false negatives. Targeted ctDNA mutations can be detected with high sensitivity and specificity using NGS on the targeted panel. Several techniques, including Tagged-Amplicon deep sequencing (TAm-seq), Safe-Sequencing System (Safe-SeqS), Cancer Personalized Profiling by deep sequencing (CAPP-Seq), and Ion Torrent, are using NGS to target panels [221].

Unique molecular identifiers (UMIs), commonly referred to as molecular barcodes, are used in digital sequencing to reduce the biases and technical noise that come with sequencing. Along with the creation of novel sequencing techniques, the idea of UMIs was first proposed twenty years ago and has since been experimentally applied. Each template DNA molecule of interest is labeled by the UMI, a random DNA sequence. Alongside the targeted sequences, the UMI is amplified during library creation. All sequence reads with identical UMI, or those belonging to the same UMI family, thus come from the same template DNA molecule after sequencing. In bioinformatics, every read in a UMI family is collapsed into a consensus read, which minimizes quantification mistakes and corrects polymerase-induced errors [222]. By adding a UMI to each barcoded DNA fragment prior to amplification, methods such as safe-SeqS have been shown to improve the sensitivity of mutation detection. In addition to lowering computational mistakes, this barcoding may successfully distinguish genuine low-frequency mutations from PCR or sequencing errors. Safe-Seqs may therefore make it possible to accurately identify uncommon tumor-specific variations, which is crucial in liquid biopsies when ctDNA is present in extremely low concentrations [223].

Advanced HCC can be profiled using ctDNA, and ctDNA NGS can be used for serial evaluation to identify alterations in the genome over time. For detecting potentially actionable gene changes and possible molecular targeted therapeutics, NGS of ctDNA offers a less invasive option. It is possible to identify dynamic changes in the molecular repertoire linked to treatment pressure in patients who are challenging to biopsy [224].

The limitations of tumor markers (mostly proteins or glycoproteins) in traditional tissue samples may be addressed by liquid biopsy. NGS technology has identified cancer-related mutations in the cfDNA of patients with different cancers, including single-nucleotide mutations, copy number alterations, methylation changes, and DNA fragmentation patterns. In addition to improving patient characterization, NGS-based analysis of cfDNA mutations can be used to guide treatment for patients with advanced cancer and for early cancer screening. It can assess several treatment alternatives in drug trials and optimize a patient’s care [225].

Organ-specific iso-miRNAs (iso-miRs) were found in a recent NGS-based analysis that evaluated several cancer types [226]. In order to demonstrate the capacity of iso-miRs to identify the organ of origin, a follow-up study created a panel of iso-miRs that successfully identified patients with triple-negative breast tumors [227]. Furthermore, it has been discovered that a number of miRNAs are RNA-edited in malignancies, and these miRNAs with altered sequences seem to acquire new biological activities [228]. Edited miR-445 promoted tumor development and metastasis in melanoma. Similarly, a limited population of altered miRNAs was found in colorectal cancer by high-throughput sequencing profiling [228,229].

## 11. Challenges and Innovations in ctDNA-Based Early Detection of HCC

cfDNA and ctDNA are documented for their potential use in the diagnosis and monitoring of the cancer. However, there is a critical issue or limitation regarding it. In early-stage cancers, ctDNA is present as a very small portion of cfDNA, and the majority of cfDNA comes from normal cells such as immune and stromal cells, especially for populations with inflammation as a common response. Therefore, very sophisticated and highly sensitive strategies are required for accurately distinguishing tumor-specific modifications. Since there is little necrosis of tumor cells in the early stages and only a limited amount of ctDNA is released into the bloodstream, cfDNA, a marker secreted into the peripheral blood by tumors, is typically not employed for screening purposes. Nonetheless, a recent study has demonstrated that ctDNA’s methylation characteristics hold significant promise for early tumor detection. After performing phase I and phase II clinical validation, researchers discovered six ideal methylated DNA markers (MDMs), including ECE1, HOXA1, cle11a, AK055957, PFKP, and EMX1. They discovered that these markers had highly specificity (92%), sensitivity (95%), and AUC (0.96) in the diagnosis of HCC [230].

Further, advanced techniques such as digital PCR and next-generation sequencing with better sensitivity and specificity have allowed a better discrimination between tumor-derived DNA and background DNA. Because of their high sensitivity, sufficient feasibility, and cost-effectiveness, biosensor-based ctDNA sensing technologies have emerged as viable substitutes for clinical ctDNA analysis. Since they can be integrated with a range of analysis techniques, have the potential to be commercially valuable, and have the desirable ability to analyze nucleic acids, ctDNA detection systems based on different biosensors (electrochemical, SPR, LSPR, or fluorescent) have drawn a lot of attention. Research on these innovative liquid biopsy and ctDNA analysis technologies has advanced significantly and has the potential to have a huge scientific impact. The specificity and reproducibility of biosensor-based ctDNA detection systems still need to be improved in order to provide more cutting-edge, workable solutions for clinical cancer therapy [231].

Further, a comparison of liquid biopsy methods reveals the advantages and disadvantages of important platforms, such as NGS, digital PCR (ddPCR), and quantitative PCR (qPCR), offering information on clinical significance and diagnostic accuracy. Precision oncology has improved tumor profiling and supported individualized therapy decisions by combining genomic, transcriptomic, and proteomic analysis with artificial intelligence (AI) techniques. Widespread clinical application is nevertheless constrained by problems such assay standardization, sample variability, regulatory complexity, and data integration, despite significant advancements. In order to enable the smooth integration of CTC-based liquid biopsy into routine cancer practice, future directions will prioritize interdisciplinary innovation, clinical validation, and strong bioinformatics frameworks. If these obstacles are overcome, liquid biopsy might be able to establish itself as a commonplace method for early cancer detection, continuous monitoring, and customized care [232].

In the future, incorporating artificial intelligence (AI) in liquid biopsy may overcome challenges such as low ctDNA yield and trouble distinguishing mutant signals from background noise and may contribute to advancing research in liquid biopsy for the treatment of HCC. Combining AI with liquid biopsy data can offer a bright future for the discovery of novel biomarkers [233]. Further, future studies may focus on the mechanisms implicated in the release of tumor-derived components in the bloodstream. It may enhance the accuracy of the diagnosis and prognosis of HCC. Moreover, it is very urgent to conduct prospective investigations to determine the best clinical use and confirm its role as a predictive tool of HCC [111].

## 12. Discussion

This review highlights the various liquid biopsy approaches, focusing on early diagnosis, prognosis, and therapeutic monitoring. Given the limitations of current diagnostic strategies like tissue biopsy and imaging tools, liquid biopsy has emerged as a minimally invasive strategy that can be used as a complement to these tools or as an advanced technique for early detection of HCC.

HCC poses a significant risk for morbidity and mortality in patients, and HCC patients have a short life expectancy. The best likelihood of recovery is when HCC is surgically removed. Regretfully, less than 20% of patients at diagnosis fit the requirements for resection. A lot of research focuses on diagnostic techniques to detect early HCC, which is determined by the tumor’s size and number of lesions [234]. Five-year survival for HCC can be increased by more than 60% with early detection and treatment. However, because HCC has a complex pathophysiology, a variable morphology, and a variety of environmental, genetic, and viral etiologies, the detection of this cancer is still difficult. For high-risk patients, current diagnostic techniques use abdominal ultrasonography with or without concurrent AFP biomarker testing [235].

Because it is widely accessible and reasonably priced, abdominal ultrasonography is the preferred imaging modality over computed tomography (CT) and magnetic resonance imaging (MRI). Abdominal ultrasonography has a 45% sensitivity for early HCC identification [236]. HCC detection rates may be increased by further screening using AFP biomarker testing. While ultrasound plus concurrent AFP testing has a lower specificity than ultrasound alone, the combination technique offers a sensitivity of about 63% for HCC identification. HCC secondary to non-viral-related etiologies has a higher AFP specificity. Therefore, while making a final selection of screening techniques, clinical evaluation is advised [235].

Significant advancements have been made, ranging from liver ultrasonography with or without contrast to dynamic multiple-phase CT and dynamic MRI with diffusion procedures. Nevertheless, a number of tumoral subtypes with varying biological behaviors have been identified by pathological, biological, genetic, and immuneochemical investigations, underscoring the necessity of reassessing conventional radiological techniques. In addition to enabling earlier diagnosis and more precise characterization (staging) of HCC lesions, CT/MRI perfusion techniques can be valuable in the prediction of survival and response to treatment. However, there are still a number of restrictions and technological issues [11]. Further, certain imaging methods, such as CT, PET, SPECT, and X-ray digital mammography, put some patients at elevated risk of ionizing radiation exposure [237].

Repeated tissue biopsies are not recommended for monitoring cancer progress or for monitoring the therapeutic response due to their invasive nature. Additionally, the heterogeneity of tumors is another important challenge associated with traditional tissue biopsy [12]. Similarly, imaging techniques, which are non-invasive and provide morphological and anatomical features of tumors, lack sensitivity for the detection of early-stage cancer and do not give information about the molecular changes in tumors [238].

Hence, there is an urgent need for novel diagnostic approaches that are less invasive, more convenient, and informative for the heterogeneity of tumors and provide real-time monitoring. Liquid biopsy appears to be promising for meeting this demand [12]. Further, liquid biopsy offers an improved diagnostic sensitivity and is easier for disease evaluation and to repeat during therapy [10]. With advanced detection strategies such as digital PCR, NGS, whole-genome sequencing, and whole-exosome sequencing, liquid biopsy is developed as a more sensitive and specific technique with enhanced diagnostic accuracy [13].

The tumor microenvironment, which is made up of pro-tumor M2 macrophages, hepatic stellate cells, hepatic dendritic cells, liver sinusoidal endothelial cells, liver resident macrophages or Kupffer cells, cytokines, fibroblasts, tumor-associated macrophages, infiltrating immune cells, and growth factors, mediates immune suppression, immune tolerance, and immune evasion, increasing tumorigenicity with improved evasion, angiogenesis, metastasis, and tumor growth [239].

A potential strategy for using the immune system to suppress tumors is cancer immunotherapy, which includes both experimental and standard-of-care treatments. Although combination treatments, especially those that combine immunotherapy with chemotherapy or radiation, have the potential to work in concert, they must be carefully managed to minimize side effects. However, factors impacting immunotherapy results, such as tumor heterogeneity, gut microbiota composition, and genomic and epigenetic changes, are advised to be considered [240]. Immunotherapy has advanced rapidly in recent years, primarily immune checkpoint inhibitors (ICIs) that target cytotoxic T-lymphocyte-associated antigen-4 (CTLA-4) and the programmed cell death-1 (PD-1) protein and its ligand, PD-L1 [206]. Various immune targeting approaches, including cancer vaccines, adoptive T-cell transfer, immune checkpoint inhibitors, and oncolytic virotherapy, have been investigated for the treatment and management of HCC. Recently, immune checkpoint inhibitor therapy and adoptive cell transfer therapy have become more and more popular, with encouraging outcomes. Nevertheless, there is still a need for a novel treatment because there is not yet a successful one. Research currently focuses on oncolytic viral therapy and combination therapy, which combines treatments like radiation, immune checkpoint therapy, adoptive cell transfer therapy, and vaccines to produce a synergistic or additive effect that boosts the liver’s immune response while having a cytotoxic effect on tumor cells [239]. Since immunotherapies such as checkpoint inhibitors, adoptive T-cell transfer, and virotherapy have been developed targeting the tumor microenvironment, liquid biopsy might be very helpful for the identification of new biomarkers, tracking therapeutic responses, and guiding precision immunotherapy in real time.

## 13. Conclusions

Despite significant advancements in cancer screening and treatment, many cancer patients still lose their lives to the disease. Traditionally, the molecular profiling of malignancies depends on surgical specimens or biopsy samples. However, even when primary tumor tissue samples are available, they are often insufficient to assign patients to the most promising treatment strategies, particularly due to the heterogeneity of the tumor lesions. Despite being invasive, needle biopsy is limited by the variability within a patient. Additionally, the use of liquid biopsy faces several challenges, including sensitivity, specificity, contamination, high cost, limited accessibility, reproducibility and reliability, and technical challenges. These challenges hinder the use of liquid biopsy. We hope that liquid biopsy will soon become essential for doctors and oncologists, representing a paradigm shift in oncology. Several recent studies have shown the prognostic significance of liquid biopsies for patients with HCC. Finding the best method to include liquid biopsy into the clinical therapy of HCC patients and adjusting current clinical practice guidelines in accordance with that finding will be the next step in the future. Additionally, it is advised to focus on the standardization of protocols for sample collection, processing, and biomarker analysis, ensuring reproducibility and clinical applicability. For the successful integration of liquid biopsy into routine clinical practice for HCC management, a multidisciplinary collaboration among oncologists, hepatologists, pathologists, and molecular diagnosticians is highly recommended.

## Figures and Tables

**Figure 1 diagnostics-15-01655-f001:**
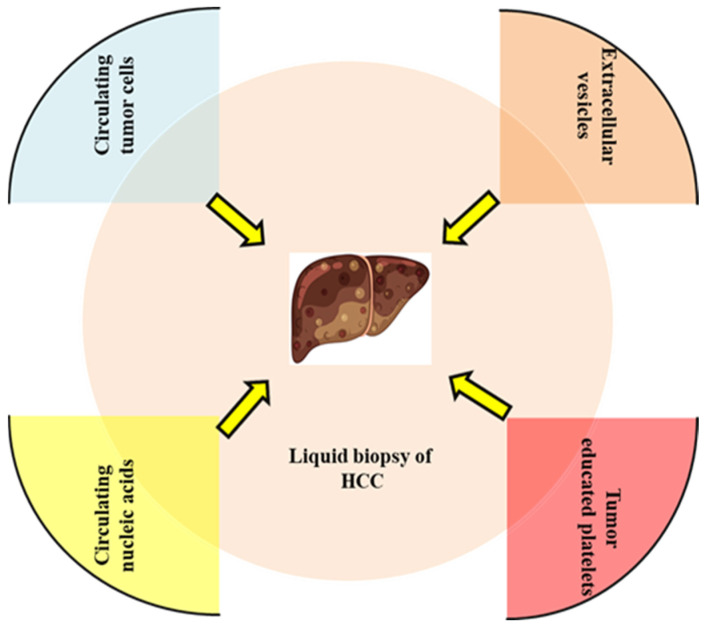
Overview of the major strategies utilized in liquid biopsy for HCC, including the analysis of CTCs, cell-free nucleic acids, and extracellular vesicles for early detection and disease monitoring.

**Figure 2 diagnostics-15-01655-f002:**
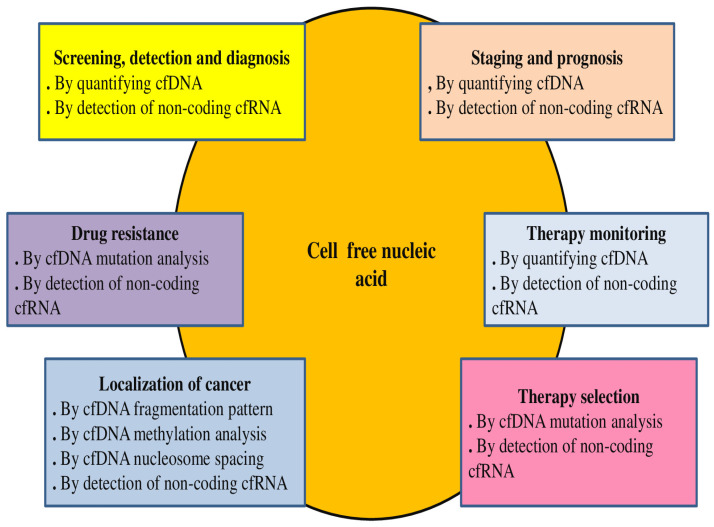
Diagnostic and therapeutic applications of cell-free nucleic acids in cancer.

**Table 2 diagnostics-15-01655-t002:** Summary of investigations using cell-free nucleic acids utilized in the detection and prognosis of various cancers.

Sample	Number of Patients	Conclusion of the Study	Cancer Type	Reference
Blood	37	In order to differentiate between benign and malignant prostate diseases, the measurement of cell-free plasma DNA may play a significant diagnostic role.	Prostate cancer	[81]
Blood	87	In cirrhosis patients, methylation indicators in cell free plasma DNA offer a novel substitute for HCC surveillance.	Hepatocelular carcinoma	[82]
Plasma from blood	85	Anti-PD-L1/VEGF therapy for u-HCC patients may benefit from pretreatment cfDNA/ctDNA analysis to predict the therapeutic outcome.	Hepatocelular carcinoma	[83]
Peripheral blood	41	Vascular invasion is linked to tumor-associated mutations found in plasma, which may be utilized to forecast a shorter recurrence-free survival period for patients with HCC. Tumor heterogeneity can be overcome by this type of biomarker.	Hepatocelular carcinoma	[84]
Blood	510	PreCar Score, a cfDNA-based screening measure for HCC in high-risk populations, was created and validated by the researchers.	Hepatocelular carcinoma	[85]
Peripheral blood	39	The methylation markers in plasma are elevated in both malignant and noncancerous tissues. MCTA-Seq will help enhance cancer diagnosis in a clinical context by facilitating the identification of ccfDNA methylation biomarkers.	Hepatocelular carcinoma	[86]
Blood	51	The promise of cell-free CP mRNA as a sensitive and selective non-invasive biomarker for HCC diagnosis is demonstrated by this study.	Hepatocelular carcinoma	[87]
Peripheral blood	14	One method for creating novel noninvasive cancer markers is the combination of plasma RNA sequencing with single-cell transcriptomic analysis.	Hepatocelular carcinoma	[88]
Peripheral blood	77	In addition to miRNAs, other ncRNA species seen in plasma small RNA sequencing can potentially function as noninvasive biomarkers. srpRNA RN7SL1 domain had dependable clinical performance for the diagnosis and prognosis of HCC.	Hepatocelular carcinoma	[89]
Blood	288	In suspected patients, the cfDNA level could not be used to distinguish breast cancer.	Breast cancer	[90]
Blood	97	The c-MET+ CTC count and cfDNA concentration may offer complementary insights into the course of the disease, underscoring the significance of integrated liquid biopsy.	Hormone receptor-positive/HER2-negative metastatic breast cancer	[91]
Peripheral blood and urine	166 (samples)	The patients with renal cell carcinoma (RCC) could be accurately classified in plasma at all stages, and urine cell-free DNA could be used to identify patients with RCC.	Renal cell carcinoma	[92]
Saliva	19	ALU115/ALU60 and ALU247/ALU60, two scfDNA integrity indices, may be used as noninvasive diagnostic biomarkers for oral squamous cell cancer.	Oral cancer	[93]
Peripheral blood	28	The cfDNA assay has the ability to detect potential targets for colorectal cancer.	Colorectal cancer	[94]
Saliva	130	Both cfDNA and cf-mtDNA demonstrated promise as precision medicine techniques for HNSCC detection.	Head and neck cancer (HNSCC).	[95]
Blood	74	Patients with advanced or metastatic pancreatic cancer may benefit from using the amounts of cfDNA generated from cancer cells as a potent biomarker for predicting the occurrence of new distant metastases.	Pancreatic cancer	[96]
Blood	153	The results lend credence to the idea that RT-ddPCR can identify cfRNA as a biomarker for solid cancer early detection.	Solid cancers	[97]
Blood	41	The results emphasize the significance of cfRNAs in clinical settings and show that they have the potential to be useful biomarkers and models for early NSCLC diagnosis.	Non-small cell lung cancer (NSCLC)	[98]
Blood	39	With a small panel of genes, the findings show a proof of principle for the use of mRNA transcripts in plasma to differentiate between precancerous situations, malignancies, and noncancerous states.	Hepatocellular carcinoma (HCC) and multiple myeloma (MM) patients	[99]
Urine	462 (Total participants)	Using small RNA-seq, a noninvasive urine extracellular vesicle miRNA-based assay may effectively and sensitively identify pancreatic cancer in its early to late stages.	Pancreatic cancer	[100]
Blood	28	The results validate cf-DNA/RNA and EV-DNA/RNA as clinically useful assays for detecting circulating tumoral HPV-DNA/RNA.	Oropharyngeal squamous cell carcinoma	[101]
Blood	140	When compared to a routine clinical workup, a noninvasive diagnostic technique for HPV + HNSCC showed increased accuracy, lower cost, and a quicker time to diagnosis.	HPV-associated head and Neck Cancer	[102]
Blood	30	When paired with common clinicopathological indicators, plasma levels of miR-923 and CA 15-3 may be utilized as a noninvasive, preoperative patient prognostic estimate.	Breast cancer	[103]
Plasma and tissue samples	35	miRNAs may be helpful prognostic or diagnostic biomarkers for the identification of cancer.	Gastric cancer	[104]
Saliva	1175	Using 270 human and microbial mRNA characteristics as markers linked to oral and throat cancer, the test performs RNA sequencing analysis.	Oral and throat cancer	[105]
Tissue and blood	45	The in-house post-surgical cfRNA showed a considerable decrease in the transcriptomic component from intestinal secretory cells. All cfRNA and tissue samples showed the same levels of HPGD, PACS1, and TDP2 expression.	Colorectal cancer	[106]
Blood	172	In the cancer group, cfRNA biomarker expression generally rose with the stage, with stage IV samples showing the highest expression.	Breast and lung cancer	[107]

**Table 3 diagnostics-15-01655-t003:** An overview of cancer diagnostic applications of extracellular vesicles.

Sample	Number of Patients	Conclusion of Study	Cancer Type	Reference
Frozen and formalin-fixed paraffin-embedded (FFPE) tissue samples	NA	Unlike EpCAM, CA9, CD70, and CD147 may be useful indicators for locating tumor-specific EVs in ccRCC.	Renal Cell Carcinoma	[164]
Blood	227	Serum-derived EV cargo may be used to enhance existing diagnostic processes and offer possible predictive and prognostic data.	Prostate cancer	[165]
Plasma	220	The EV signature can be used as an independent predictive factor for progression-free survival in metastatic breast cancer patients receiving treatments, and it can reliably track the response to treatment across training, validation, and prospective cohorts.	Breast cancer	[166]
Colon cancer cell line	NA	With 75.7% sensitivity, it was discovered that plasma EVs from colon cancer patients had higher levels of tetraspanin 1 than those from healthy controls	Colon cancer	[167]
Peripheral blood	159	Small EVs in circulation colon cancer patients’ enriched fractions have a unique miRNA profile, and miRNA from small EVs may be a useful diagnostic for early CC detection.	Colon cancer	[168]
Gastric juice	NA	The presence of EVs in isolates from the gastric juice (GC) of patients with gastric cancer suggests that GJ-EVs have a partial impact on their microenvironments and that GC patients’ GJ-EVs will aid in the pathophysiology of GC.	Gastric cancer	[169]
Peripheral blood mononuclear cells (PBMCs) or ascites cells	137	A useful diagnostic tool for liver cancer is the separation and detection of plasma LC3B+ EVs harboring bioactive compounds. These EVs may potentially be employed as a possible marker for immunological monitoring and clinical prognostication.	Liver cancer	[130]
Venous blood	40	Disease activity and treatment response are correlated with the induction of apoptosis in CD8+ T cells by sEV.	Head and neck cancer	[170]

**Table 4 diagnostics-15-01655-t004:** Comparative summary of liquid biopsy and tissue biopsy.

Parameter	Tissue Biopsy	Liquid Biopsy	Reference
Invasiveness	More invasive	Minimal or non-invasive	[5,26]
Monitoring tumor heterogeneity	Limited	Broad	[25,30]
Time-to-result	Longer due to tissue processing and histopathology	Faster	[27,28]
Real time disease progress monitoring	Painful, risky, and impractical	Excellent for monitoring disease progression and therapeutic responses	[20,26]
Cost	Higher for imaging/surgery and for repeated sampling	Varies with platform and is cost-effective for repeated sampling	[5,25]
Sample stability	Stable after once processed	Sensitive to handling	[204]
Molecular profiling	High resolution	Can track genetic, transcriptomic, and epigenetic changes	[10,201,203]
Clinical utility	Gold standard for initial diagnosis	Complementary for prognosis, therapy monitoring, and detecting resistance mechanisms	[26]
Access to tumor tissue	Direct access to tumor tissue is required	No direct access to tumor tissue is required	[67]

## Data Availability

During the study, no new data were created.

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
