# Peer review of "Role of Liquid Biopsy for Early Detection, Prognosis, and Therapeutic Monitoring of Hepatocellular Carcinoma"

_diagnostics, 2025, doi:10.3390/diagnostics15131655_

Round 1

Reviewer 1 Report

Comments and Suggestions for Authors

This manuscript highlighted the potential of Liquid biopsy for diagnostic and prognostic strategy as a minimally invasive and promising alternative to traditional tissue biopsy. I have few comments for major revision:

Author needs to highlight the method of doing liquid biopsy in the abstract section.

In introduction, the traditional tissue biopsy literature is missing. 

Why there is a need of liquid biopsy?

Why this technique could be important as compared to available imaging modalities.

The discussion section is missing in this review. The flow of review is lacking. The discussion of the current situation in tissue biopsy, imaging and requirement of liquid biopsy in diagnosis and prognosis is missing. 

Author Response

This manuscript highlighted the potential of Liquid biopsy for diagnostic and prognostic strategy as a minimally invasive and promising alternative to traditional tissue biopsy. I have few comments for major revision:

  1. Author needs to highlight the method of doing liquid biopsy in the abstract section.

Response- Thank you for this insightful comment, I have revised the abstract according to your valuable suggestions.

  1. In introduction, the traditional tissue biopsy literature is missing.

Response- Thank you for this insightful comment, I added traditional tissue biopsy literature in the introduction section. Additionally, I added in discussion also.

  1. Why there is a need of liquid biopsy?

Response- Thank you for your valuable comment, I discussed it in the discussion section.

4. Why this technique could be important as compared to available imaging modalities.

Response- We are sincerely grateful to the reviewer for this insightful comment. I added a dedicated Discussion section (Section 13) that comprehensively addresses the current limitations of tissue biopsy and conventional imaging modalities in hepatocellular carcinoma (HCC) management. We have also elaborated on the growing clinical need for liquid biopsy in both diagnostic and prognostic contexts.

5. The discussion section is missing in this review. The flow of review is lacking. The discussion of the current situation in tissue biopsy, imaging and requirement of liquid biopsy in diagnosis and prognosis is missing.

Response- I am sincerely grateful to the reviewer for this insightful comment. I added a dedicated Discussion section (Section 13) that comprehensively addresses the current limitations of tissue biopsy and conventional imaging modalities in hepatocellular carcinoma (HCC) management. I have also elaborated on the growing clinical need for liquid biopsy in both diagnostic and prognostic contexts. Furthermore, to enhance coherence and improve the logical flow, we revised several sections of the manuscript to better integrate the clinical context of HCC, highlight diagnostic challenges, and clarify how liquid biopsy addresses unmet clinical needs. These revisions collectively provide a clearer rationale for the importance of liquid biopsy as a complementary—and in some cases superior—approach compared to traditional modalities.

Reviewer 2 Report

Comments and Suggestions for Authors

This review provides an overview of current and emerging liquid biopsy strategies in HCC, with a focus on their applications in early detection, prognosis, and monitoring treatment response.

However, various immune-targeting strategies—such as adoptive T-cell transfer, cancer vaccines, and virotherapy—are under active development, aiming to modulate the HCC immune microenvironment. These approaches target distinct cellular players and pathways, expanding the landscape of available immunotherapies and highlighting potential future modalities.

Author Response

This review provides an overview of current and emerging liquid biopsy strategies in HCC, with a focus on their applications in early detection, prognosis, and monitoring treatment response.

However, various immune-targeting strategies—such as adoptive T-cell transfer, cancer vaccines, and virotherapy—are under active development, aiming to modulate the HCC immune microenvironment. These approaches target distinct cellular players and pathways, expanding the landscape of available immunotherapies and highlighting potential future modalities.

Response- I am sincerely grateful to the reviewer for this insightful comment. I added an expanded discussion section addressing the role of immune-targeting approaches and highlighted key modalities such as immune checkpoint inhibitors, adoptive T-cell transfer, cancer vaccines, and oncolytic virotherapy, and their potential to modulate the tumor immune microenvironment. Additionally, I emphasized complementary role of liquid biopsy for tracking tumor stages, treatment responses, and detecting immune escape mechanisms, which reflects its clinical relevance in precision immunotherapy. 

Reviewer 3 Report

Comments and Suggestions for Authors

This proposed review provides a comprehensive overview of liquid biopsy as an emerging tool for the management of Hepatocellular Carcinoma (HCC). It details various minimally invasive techniques that analyze tumor-derived materials in bodily fluids. The authors highlight the potential of these methods for early HCC detection, prognosis, and monitoring treatment response, positioning them as a promising alternative to invasive tissue biopsies. However, the manuscript also emphasizes that significant challenges related to sensitivity, standardization, and cost currently limit their routine clinical use. The review concludes that future progress depends on standardizing protocols and fostering multidisciplinary collaboration to fully integrate liquid biopsy into HCC patient care. Despite its overall good quality of content, there are some areas that could be better addressed. Here is a complete analysis and suggestions to the authors.

  1. The abstract states that "multi-analyte liquid biopsy panels showed encouraging results". This is a key concept in the field. To improve, the abstract could briefly mention whythis is the case (e.g., by combining markers like ctDNA and proteins to increase sensitivity) to better capture the cutting-edge nature of the research.
  2. The introduction mentions that conventional diagnostics like imaging are insufficient for early-stage tumors and that liquid biopsy is anticipated to play a key role. To strengthen the introduction, these two points could be more directly linked, creating a clear problem-solution narrative from the outset.
  3. The citation supporting the future role of liquid biopsy in HCC is a study focused on colorectal cancer. While acceptable for a general point, finding a citation more specific to HCC would strengthen the claim.
  4. Regarding CTCs, the manuscript states that EpCAM is a popular marker but its expression varies. This section could be improved by delving deeper into the clinical implications of this variability—specifically, that EpCAM-based enrichment may fail to capture the most aggressive CTCs that have undergone EMT.
  5. The review mentions that mesenchymal and hybrid CTCs are linked to poor prognosis. It would be more impactful to discuss the challenges of standardizing the definition of these phenotypes for clinical use. For example, what specific thresholds for vimentin or twist expression define a "mesenchymal" CTC?
  6. Table 1 is meant to show the significance of CTCs. However, it includes only one study on HCC , with the rest focusing on lung , prostate, and other cancers. The table should be revised to feature more HCC-specific studies to better support the review's main topic.
  7. Similar to Table 1, Table 2 includes only two HCC-specific studies. To better serve the manuscript's focus, the table should be populated with more examples from HCC literature.
  8. The review discusses various methods for isolating exosomes, such as ultracentrifugation and immuno-affinity capture. A key area for improvement would be to explicitly state that the lack of a standardized, universally accepted isolation protocol is a major barrier to the clinical translation of EV-based diagnostics. This variability makes it difficult to compare results across studies.
  9. Regarding the current management of HCC and its link to liquid biopsy, this section currently stands alone. Its value to the review would be immensely increased if each treatment modality mentioned was explicitly linked back to a potential application of liquid biopsy. For example, after discussing sorafenib, the author could add, "The emergence of sorafenib resistance could be monitored non-invasively by tracking specific mutations in ctDNA or resistance-conferring miRNAs in EVs."
  10. Regarding the use of liquid biopsy as a contemporary tool to tissue biopsy, to make the comparison more direct and visually accessible, a summary table could be created. This table could list parameters (e.g., Invasiveness, Capturing Tumor Heterogeneity, Cost, Turnaround Time, Monitoring Capability) and compare liquid biopsy vs. tissue biopsy for each.
  11. The review mentions that techniques like Safe-SeqS boost sensitivity but doesn't explain how. Adding a brief explanation (e.g., "by using unique molecular identifiers to barcode original DNA fragments, allowing for computational error correction and the confident detection of rare variants") would provide greater technical depth.
  12. Please consider the emerging role of non-invasive serum markers and scores that have been largely validated for early cancer screening and detection. Please refer to PMID 36561127, PMID 37852989, PMID 35589256, etc.
  13. Regarding the paragraph for source of ssDNA, this single paragraph should be removed as a standalone section. The information should be integrated into Section 4.1 on ctDNA. The use of "ssDNA" should also be harmonized with "cfDNA/ctDNA" used throughout the manuscript to avoid confusion.

Author Response

  1. The abstract states that "multi-analyte liquid biopsy panels showed encouraging results". This is a key concept in the field. To improve, the abstract could briefly mention why this is the case (e.g., by combining markers like ctDNA and proteins to increase sensitivity) to better capture the cutting-edge nature of the research.

Response- Thank you for this insightful comment, I have revised the abstract according to your valuable suggestions.

2. The introduction mentions that conventional diagnostics like imaging are insufficient for early-stage tumors and that liquid biopsy is anticipated to play a key role. To strengthen the introduction, these two points could be more directly linked, creating a clear problem-solution narrative from the outset.

Response- We thank the reviewer for the valuable suggestion. A concise paragraph has been added in the introduction to directly link the limitations of conventional imaging with the emerging role of liquid biopsy. This link has been further elaborated in the discussion section to strengthen the overall narrative.

  1. The citation supporting the future role of liquid biopsy in HCC is a study focused on colorectal cancer. While acceptable for a general point, finding a citation more specific to HCC would strengthen the claim.

Response- I am sincerely grateful to the reviewer for this insightful comment. I added two paragraph in the “Challenges and Innovations in ctDNA-Based Early Detection of HCC” section and highlighted them.

4. Regarding CTCs, the manuscript states that EpCAM is a popular marker but its expression varies. This section could be improved by delving deeper into the clinical implications of this variability—specifically, that EpCAM-based enrichment may fail to capture the most aggressive CTCs that have undergone EMT.

Response- We sincerely thank the reviewer for this insightful comment. In response, we have expanded the section on circulating tumor cells (CTCs) to discuss the clinical implications of EpCAM variability.

5. The review mentions that mesenchymal and hybrid CTCs are linked to poor prognosis. It would be more impactful to discuss the challenges of standardizing the definition of these phenotypes for clinical use. For example, what specific thresholds for vimentin or twist expression define a "mesenchymal" CTC?

Response- We sincerely thank the reviewer for this insightful comment. In response, we have expanded the section on circulating tumor cells (CTCs) to discuss the challenges of standardizing the definition of these phenotypes for clinical use, and specific thresholds for vimentin or twist expression.

  1. Table 1 is meant to show the significance of CTCs. However, it includes only one study on HCC , with the rest focusing on lung , prostate, and other cancers. The table should be revised to feature more HCC-specific studies to better support the review's main topic.

Response- I am sincerely grateful to the reviewer for this insightful comment. I revised the table and added HCC-specific studies.

7. Similar to Table 1, Table 2 includes only two HCC-specific studies. To better serve the manuscript's focus, the table should be populated with more examples from HCC literature.

Response- I am sincerely grateful to the reviewer for this insightful comment. I revised the table and added HCC-specific studies.

8. The review discusses various methods for isolating exosomes, such as ultracentrifugation and immuno-affinity capture. A key area for improvement would be to explicitly state that the lack of a standardized, universally accepted isolation protocol is a major barrier to the clinical translation of EV-based diagnostics. This variability makes it difficult to compare results across studies.

Response- I am sincerely grateful to the reviewer for this insightful comment. I added the a paragraph in section “Extracellular vesicles or exosomes: Implications for cancer progression and biomarker discovery”

9. Regarding the current management of HCC and its link to liquid biopsy, this section currently stands alone. Its value to the review would be immensely increased if each treatment modality mentioned was explicitly linked back to a potential application of liquid biopsy. For example, after discussing sorafenib, the author could add, "The emergence of sorafenib resistance could be monitored non-invasively by tracking specific mutations in ctDNA or resistance-conferring miRNAs in EVs."

Response- I am sincerely grateful to the reviewer for this insightful comment. I added a paragraph in section “An overview of current management of HCC.”

10. Regarding the use of liquid biopsy as a contemporary tool to tissue biopsy, to make the comparison more direct and visually accessible, a summary table could be created. This table could list parameters (e.g., Invasiveness, Capturing Tumor Heterogeneity, Cost, Turnaround Time, Monitoring Capability) and compare liquid biopsy vs. tissue biopsy for each.

Response- I am sincerely grateful to the reviewer for this insightful comment. A comparative summary of liquid biopsy and tissue biopsy is provided in table 3 in section “Liquid biopsy as a contemporary tool to tissue biopsy.

11. The review mentions that techniques like Safe-SeqS boost sensitivity but doesn't explain how. Adding a brief explanation (e.g., "by using unique molecular identifiers to barcode original DNA fragments, allowing for computational error correction and the confident detection of rare variants") would provide greater technical depth.

Response- I am sincerely grateful to the reviewer for this insightful comment. A paragraph has been added in the section “Next generation sequencing and liquid biopsy”

12. Please consider the emerging role of non-invasive serum markers and scores that have been largely validated for early cancer screening and detection. Please refer to PMID 36561127, PMID 37852989, PMID 35589256, etc.

Response- I am sincerely grateful to the reviewer for this insightful comment. A paragraph is added in the section “An overview of current management of HCC” and the recommended references are cited.

13. Regarding the paragraph for source of ssDNA, this single paragraph should be removed as a standalone section. The information should be integrated into Section 4.1 on ctDNA. The use of "ssDNA" should also be harmonized with "cfDNA/ctDNA" used throughout the manuscript to avoid confusion.

Response- We sincerely thank the reviewer for this insightful comment. We revised the first paragraph of section 4.1. We integrated this paragraph for source of ssDNA at the end of section 4.1.

Round 2

Reviewer 3 Report

Comments and Suggestions for Authors

Authors addressed required amendments correctly.